# Heat exchange characteristics of underground and pavement buried pipes for bridge deck heating conditions

**Xuegui Zheng** [1] *, **Zhongbo Song**[2], **Yanping Ding**[3]

**1** Chongqing Vocational Institute of Engineering, Chongqing, China, **2** China Railway Construction Bridge Engineering Bureau Group CO., LTD, Tianjin, China, **3** Chongqing Jiaotong University, Chongqing, China

* zhengxuegui163@163.com

## Abstract

Geothermal energy is increasingly employed across diverse applications, with bridge deck snow melting emerging as a notable utilization scenario. In Jinan city, China, a project is underway to utilize ground source heat pumps (GSHPS) for heating bridges. However, essential operational parameters, including fluid medium, temperature, and heat exchange details, are currently lacking. This study addresses the thermal design challenges associated with ground heat exchangers (GHE) for bridge heating through a combination of numerical modeling and field experiments. Utilizing software Fluent, a refined three-dimensional multi-condition heat transfer numerical analysis was carried out. Field tests based on actual operating conditions were also conducted and the design parameters were verified. The results indicate that an inlet temperature of 5°C and an aqueous solution of ethylene glycol with a mass concentration of 35% as the heat exchange medium are suitable for the GSHPS in Jinan; Moreover, the influence of backfill material and operation time on the heat transfer efficiency was revealed and the suitable material with 10% bentonite and 90% $SiO_2$ was suggested; Finally, based on the influence of the pipe spacing on the heating characteristics of bridge deck, the transition spacing of 0.2 m is given for the temperature response of the bridge deck. This comprehensive study contributes valuable insights through simulation and experimental analysis of the thermal environment variation, aiming to advance the development of GSHPS for bridge deck heating in Jinan, China.

## 1 Introduction

During winter, over 70% of roads in China experience ice and snow conditions, leading to significant challenges that impact the safety, mobility, and productivity of the transportation system [1]. Snow accumulation can pose serious threats, causing reduction of the adhesion coefficient between vehicle tires and the road surface. This reduction contributes to issues such as vehicle slippage, deviation, and extended braking distances, posing risks to the personal safety of drivers and passengers. In the United States alone, each year witnesses over 1300 fatalities and more than 116,800 injuries resulting from vehicle crashes on snowy, slushy,

**Data Availability Statement:** All relevant data are within the manuscript.

**Funding:** The first author thanks for funding from Chongqing Vocational Institute of Engineering (No. 97682DZ); The funders had no role in study

design, data collection and analysis, decision to publish, or preparation of the manuscript.

**Competing interests:** The authors have declared that no competing interests exist.

or icy pavement [2]. Over $6 and $10 billion are spent on the snow and ice removal annually in the USA and China, respectively. This cost has continued to increase in recent years [3].

Currently, passive methods, including mechanical and chemical snow melting approaches, are primarily employed for ice and snow removal [4, 5]. Mechanical methods often involve lane closures or traffic disruption, leading to inefficient snow removal and incomplete clearance. Additionally, snow removal agents have the potential to corrode steel bars in cement concrete pavement or bridge structures, impacting road and bridge durability. Moreover, these agents contribute to environmental pollution when discharged into water and soil. The reliance on passive measures falls short of meeting the demands for green, energy-saving, environmentally friendly, and highly efficient solutions [4]. The thermal snow melting method is recognized as an active, environmentally friendly technology that harnesses heat from various sources [3], including hot water, solar energy, geothermal energy, and electricity, to facilitate snow melting. This method encompasses cable heating, conductive concrete heating, and circulating thermal fluid heating technologies. Cable heating and conductive concrete heating directly rely on electrical energy, offering stability but exhibiting limitations in energy utilization efficiency.

As an effective and green energy source, geothermal energy is gradually used in a variety of building facilities of the world [6–8]. The thermal energy from the ground can be utilized as a heat source and heat sink in heat pumps to reduce the rapid consumption of non-clean energy and increase the electricity energy utilization efficiency. The systems of ground source heat pumps (GSHPS), utilizing the geothermal energy are known as geothermal heating pumps that absorb/transfer thermal energy to/from the earth using a series of buried pipes and which can be considered one of the most efficient technologies in the heating/cooling industry. Over the last few decades, GSHPS, because of the high-performance factor and more economical operating costs, have acquired significant attention as sustainable alternative energy resources for building cooling/heating applications [9]. The first interest in the technology of GSHPS was known after the Second World War until the early 1950s in North America and Europe. Another significant surge occurred in the 1970s following the global oil crisis, marking a period of increased GSHPS activity worldwide. As the core heat extraction part of the GSHPS, the ground and pavement heat exchangers play an important role in determining the energy efficiency of the entire system operation [10–12]. Many factors affect the thermal performance of ground heat exchanger (GHE), including the type of buried pipes, the backfill materials, the inlet and outlet temperatures, and also the depth of GHE. Thus, several studies have been conducted from different perspectives. Ajarostaghi et al. [13] studied the performance of a single U-shaped vertical buried pipe heat exchanger by numerical simulation. The results show that the inlet volume flow, the inlet fluid temperature, and the backfill material's thermal conductivity significantly impact the thermal performance of the buried pipe heat exchanger. The outlet fluid temperature is significantly reduced by reducing the inlet volume flow and increasing the thermal conductivity of the backfill material and the inlet fluid temperature. Javadi et al. [14] reviewed various backfill materials (e.g. bentonite, silica sand and coarse/fine sand) and their effects on the performance of buried pipe heat exchangers. Shi et al. [15] comprehensively analyzed the impact of many key factors on GHE performance by comparing numerical simulation and grey correlation analysis. The results show that the most significant factor affecting the performance of GHE is groundwater flow rate, circulating flow rate, precipitation and air temperature. In addition, Basok et al. [16] used a numerical simulation method to study the influence of the filtration characteristics of soil as porous media on the operation characteristics of the soil heat exchanger, and determined the dependence of the energy characteristics of soil heat exchanger and heat pump operation on the porous media, the porosity and particle size of soil. Tang et al. [17] established a comprehensive numerical simulation model to

evaluate the annual performance of shallow-buried hole heat exchangers installed in different soils. The results showed that the performance of shallow-buried hole heat exchangers installed in sandy soil was 8% higher than that of clay. The same performance difference between sand and clay was observed in most survey scenarios. Changes in meteorological conditions, the thermal conductivity of grouting, and thermal load level will affect the annual average heat pump performance. Meng et al. [18] established the simulation of GHEs under typical geological conditions and studied the soil freezing characteristics and dynamic performance of GHEs under different seepage velocities. Zhang et al. [19] built a three-dimensional simulation platform considering heat and moisture transfer, seepage, and freezing. They simulated the impact of the arrangement of the tube banks and the type of GHE on the ground source heat pump.

Using GSHPS for bridge deck heating is one of the new application of geothermal utilization in recent years [2, 20–26]. Han et al. [2] provided a relatively simple framework that combines overall energy analyses and computational model to estimate the technical and economic viability of GHE pile based snow melting system for bridge deck in the United States with different climatic conditions. Liu et al. [27] developed a transient snow melting model which evaluates the heat flux required for instantaneous snow melting based on typical Canadian weather. They found that The heating loads are determined by local weather conditions including snowfall rate, air temperature, wind speed and solar radiation. Chen et al. [28] investigated the thermal performance of a bridge deck with a hydronic heating system under heating conditions based on field tests and suggested that the inlet fluid temperature should be controlled above 2.3 times the absolute value of the air temperature to keep the slab surface temperature above 0˚C. For existing bridges, Yu et al. [29] developed a new attached hydronic loop design for geothermal heating of bridge decks and conducted feasibility test. Numerical methods were adopted to study the melting process and predict the surface conditions of a bridge [30, 31]. Previous studies concluded the inlet temperature and flow rate of medium, the thermal conductivity of backfill material, the borehole depth and spacing having different effects on the engineering performance of GSHPS. Although some experimental and simulation analysis on GSHPS had been conducted, theoretical guidance on the design of GHE with the bridge deck heating conditions has not been unified.

Jinan (116˚E, 37˚N), capital of Shandong Provence of China, is located in a hilly area, with several highways passing through the area and multiple bridges constructed. To reduce the impact of snowfall on transportation in winter, it is planned to eliminate snow accumulation through GSHPS. To further improve the geothermal utilization in heating system for bridge decks under the specific geological and atmospheric conditions for Jinan, it is crucial to study the heat transfer capacity of the heat exchangers and obtain the optimal parameters governing its performance. The study focuses on the heat exchange characteristics of underground and pavement buried pipes with various parameters for an actual bridge deck heating system conducted in Jinan, China. The influence of different inlet temperatures of buried pipes, backfill material ratios and other factors on heat transfer efficiency are investigated through simulation and field test study, which can be used to guide the design and operation of the system.

## 2 Thermal design of GHE for bridge deck heating and modeling

### 2.1 The basic characteristics of the project

The geotechnical temperature and physical parameters around the heat exchanger were obtained from borehole interpretation data for winter surveys. On average, the local ground temperature in the shallow layer above 100 m is about 15˚C. The thermophysical parameters of the geotechnical body around the buried pipe are the weighted average values of the geotechnical data from different layers, with the density of the geotechnical body being 1900 kg/

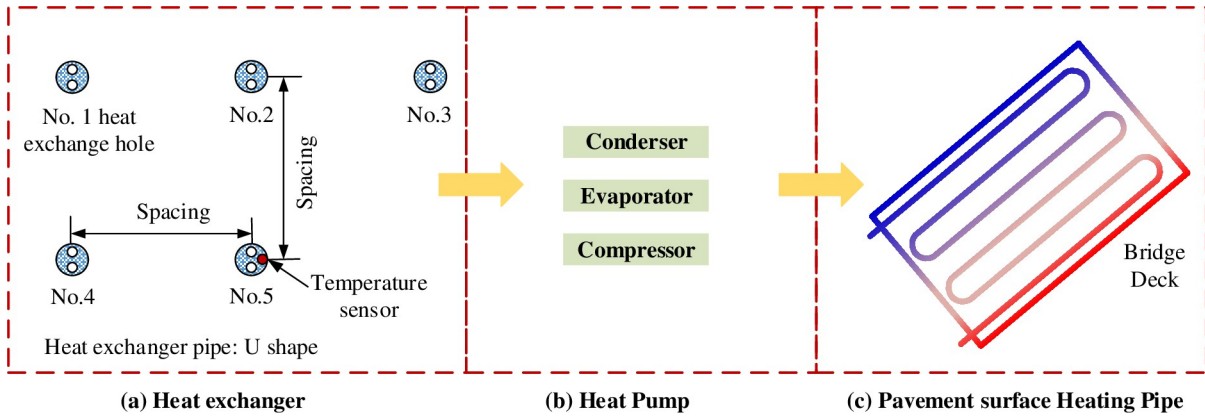

**(a) Heat exchanger**　　　　**(b) Heat Pump**　　　　**(c) Pavement surface Heating Pipe**

**Fig 1. Schematic diagram of the GSHPS for bridge heating.**

m³, the specific heat capacity being 1820 J/(kg·K), and the thermal conductivity being 1.8 W/ (m·K). The design process are as follows:

(1) When the GSHPS is used for bridge deck heating as shown in Fig 1, its load mainly includes raising snow temperature, snow melting latent heat load, convection heat transfers between snow layer and atmosphere, long wave radiation between snow layer and sky, and solar radiation. Utilizing the environmental parameters specific to the Jinan region and aggregating the sub-loads related to snowmelt, the total snowmelt load is calculated as 235.49 W/m², with detailed information provided in the S1 File.

S1 File of the total snowmelt load(2) Based on the bridge deck area and the power characteristics of the heat pump, the total heat exchange of the buried pipe can be calculated to be about 32 kW. The heat extraction rate of underground heat exchangers in Jinan area is about 40 W/m, resulting in a tube length of 800 m for underground heat exchangers. This project adopts U-type buried pipes, so when using 5 buried pipes, the minimum depth of each pipe is 80 m.

(3) The heat pump host (YCFW050, YORK company) was selected for the bridge deck heating system. According to its standard application requirements in heat extraction working conditions, the heat source side water flow rate is specified as 10.97 m³/h. Since each host heat source will be connected to a 5-way branch port size DN40 manifold. It is easy to calculate the buried pipe inlet flow rate of 0.7 m/s.

## 2.2 Three-dimensional heating transfer model

To obtain more accurate engineering design parameters, numerical model analysis under multiple working conditions was carried out, which involves matching different material and physical parameters and necessary initial boundary constraints to the model. The influence of different performance parameters on heat extraction capacity under the operating conditions of the whole snow melting system was analyzed. The factors that considered include the inlet medium temperature and flow rate of the heat exchanger; varying concentrations of the heat transfer media, different types of backfill materials, varying borehole depths and single-hole spacing.

The heat exchange process between an underground heat exchanger and the surrounding soil was numerically studied by using computational fluid dynamics (CFD) software FLUENT. This method allows for clear visualization of the temperature change trends in the soil in the

depth, as well as the temperature field changes around the heat exchanger. To ensure the accuracy and reliability of the simulation calculation results, a three-dimensional model built replicates the geometry of the actual project, as illustrated in Fig 2. The model comprises the following components: heat transfer medium in the pipe, a single U-shaped pipe wall, backfill area, and soil area. Through reliability verification, the model can be well used to simulate the temperature field analysis of GSHPS. The backfill area and fluid domains are meshed with sufficient density to meet the calculation accuracy requirements with the aid of ICEM software [32], while also accounting for the viscous effect of the fluid on the inner wall of the tube.

The governing equations for the non-isothermal pipe flow numerical model are described as follows:

$$\frac{\partial \rho_w}{\partial t} + \text{div}(\rho_w u) = 0 \tag{1}$$

$$\frac{\partial(\rho_w u)}{\partial t} + \text{div}(\rho_w u) = \text{div}(\mu \text{grad} u) - \text{div}(P) + F \tag{2}$$

$$\frac{\partial(A\rho_w T)}{\partial t} + \text{div}(A\rho_w uT) = \text{div}\left(A\frac{k}{C_p}\text{grad}T\right) + S_T \tag{3}$$

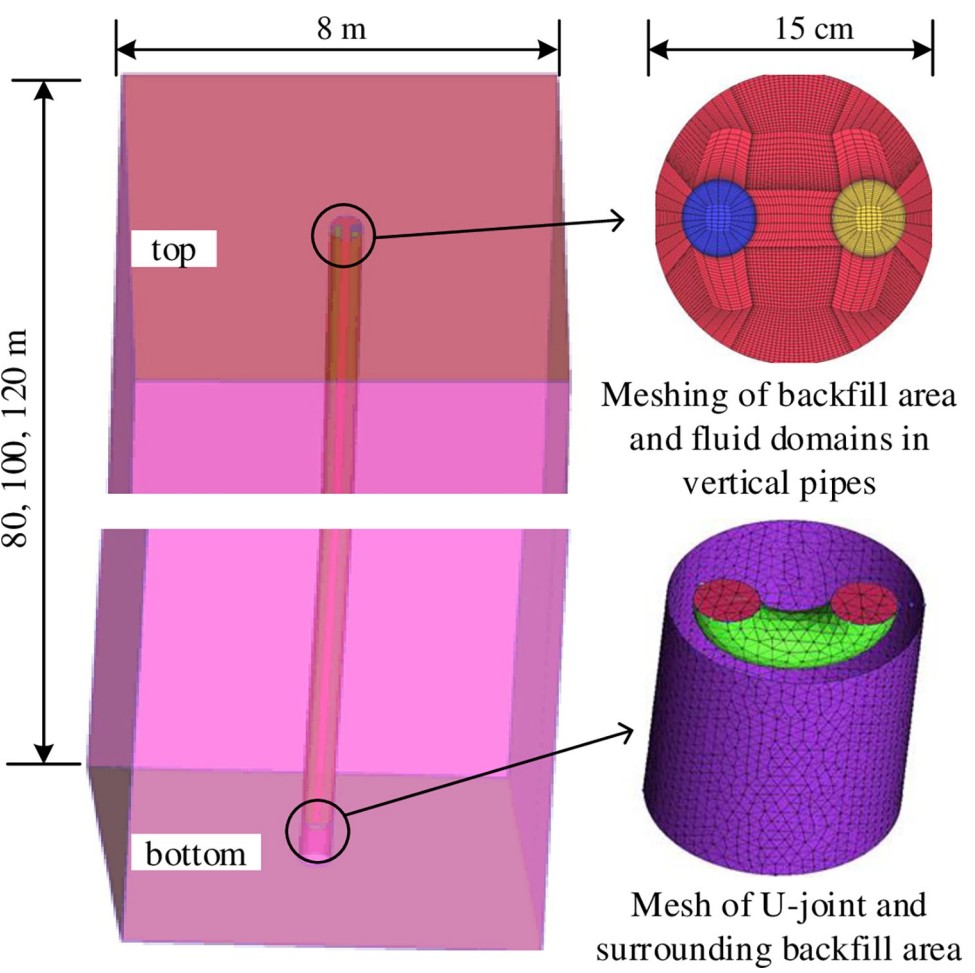

**Fig 2. Three-dimensional geometric model and the meshing of buried U-type pipe.**

**Table 1. Boundary definitions.**

| Boundary Areas | Setting Type | Setting Instructions |
|---|---|---|
| inlet | Velocity-inlet | Buried pipe inlet medium temperature, flow rate |
| Outlet | Pressure-outlet | Buried pipe outlet medium absolute outflow pressure |
| Model Top | Wall | The upper boundary wall thermostatic temperature |
| Model bottom | Wall | Lower boundary wall thermostatic temperature |
| Soil zone peripheral surface | Wall | The constant temperature at the outer boundary wall |
| Media in the pipe | Fluid | Physical parameters of heat exchange flow medium |
| Backfill area | Solid | Backfill material physical parameters |
| Soil zone outside the borehole | Solid | Soil properties |

where, $\rho_w$ is the density of the water, kg/m$^3$. $u$ is the fluid velocity, m/s. $P$ is the pressure in N/m$^2$. $\mu$ is the viscosity of the fluid, Pa·s. $F$ represents the volume force in N/m$^3$. $C_p$ is the heat capacity at constant pressure in J/(kg·K). $T$ is the temperature in Kelvin. $k$ is the thermal conductivity in W/(m·K), $S_T$ is a source/sink term accounting for heat exchange with the surroundings through the pipe wall in W/m. $A$ is the cross section area of the pipe in m$^2$. Eqs (1)–(3) describes the heat transfer in the circulating fluid, encompassing conduction, convection, and dissipation, as well as the heat transfer between circulating fluid and the pipe wall (via the source/sink term $S_T$).

The heat transfer process between the ground heat exchanger and the surrounding soil is a non-stationary process that involves multi-layer heat transfer media and convection heat transfer. To accommodate the intricate geometry of the 3D buried pipe heat exchanger and the limited operating time of heating systems, specific assumptions and simplifications are introduced to uphold the accuracy of the established model. These assumptions include: (1) Considering the soil as a solid material; (2) Neglecting the external weather and shallow surface heat exchange, thermal convection, and thermal radiation; (3) Disregarding external weather, shallow surface heat exchange, thermal convection, and thermal radiation.

Table 1 shows the specific boundary definitions and settings. The buried pipe's inlet and outlet are defined as velocity inlet and free outflow, respectively. The subsequent calculations for different conditions include the inlet temperature and flow velocity. The top and bottom surfaces of the model, as well as the distal boundary surfaces on the radial direction of the model, are defined as wall surfaces.

# 3 In situ study description

## 3.1 Construction of the system

The bridge is 28 m in length and 25 m in width (four car lanes and two sidewalks). From top to bottom, the bridge deck structure consists of a 10 cm thick asphalt waterproof layer, a 15 cm thick cast-in-place concrete layer, and an 80cm thick precast hollow concrete slab. The construction process of the bridge snow melting project involves several steps, including drilling underground pipes, laying underground and bridge deck pipelines, collecting pipelines, and connecting heat pumps. Holes are drilled into the slope of the bridge head to a depth of 100 m. To study the effect of pipe spacing on bridge deck temperature, three different pipe spacing schemes of 0.15 m, 0.2 m, and 0.25 m were implemented. The heat exchange pipes were tied to the reinforced mesh of the cast-in-place concrete layer and buried 2 cm below the reinforced cement concrete surface. As a result, the pipes were 12 cm beneath the bridge deck surface. The heat exchange pipes beneath the asphalt layer and within the bridge deck were collected and connected to the heat pump system. All exposed pipelines are insulated using

**Table 2. Physical parameters of ethylene glycol aqueous solution with different concentrations.**

| Mass concentration | Freezing point | Viscosity | Density | Specific heat capacity | Thermal conductivity |
|---|---|---|---|---|---|
| | (100.7kPa)°C | mPa·s | kg/m$^3$ | J/kg·K | W/m·K |
| 25% | -10.7 | 2.84 | 1038 | 3740 | 0.464 |
| 30% | -14.1 | 3.28 | 1047 | 3640 | 0.444 |
| 35% | -17.9 | 3.83 | 1054 | 3550 | 0.436 |
| 40% | -22.3 | 4.47 | 1061 | 3460 | 0.409 |

prefabricated polyurethane foam technology. Unlike conventional methods for measuring fluid temperature, this test focuses on capturing the temperature variation characteristics of backfill materials during the operational period of the bridge heating system. A temperature measurement line was installed on-site along with the No. 5 buried pipe. The positions of the temperature measurement holes are indicated in Fig 1.

### 3.2 Field test procedure

Research has shown that the medium concentration of heat pump circulating fluid has little effect on the heat exchange rate. Table 2 shows the physical parameters of ethylene glycol aqueous solutions with different mass concentrations. For every 5% difference in mass concentration, the solution has a significant difference in freezing point and viscosity, while the difference in thermal physical parameters is small. In the field test, an aqueous solution of ethylene glycol with a mass concentration of 35% as the heat exchange liquid medium. To assess the operational efficiency of the buried pipe bridge deck heating system, the heat pump was activated with a power of 40 kW to circulate the liquid. The inlet medium temperature was set at 5°C, and the flow rate during operation was controlled at 0.7 m/s. The testing process extended for 3 hours, and to mitigate the impact of environmental temperature on the system, the testing time was selected in the morning, from 6 am to 9 am on January 17, 2023. The essential temperature measurement equipment comprises an AC voltage regulator, a resistance thermometer sensor, field data acquisition, and a data adapter. The resistance thermometer sensor has a measurement range of -50 to 150°C with an accuracy of 0.05°C. The field data acquisition is employed to gather temperature signals, which are then connected to the data adapter. The monitoring system automatically stores the temperature data obtained in the computer system. The temperature changes throughout the day at the site are depicted in Fig 3. The ambient temperature during testing was approximately minus 2°C. Temperature measuring sensors generated 1 temperature data point per 5 meters of length.

## 4 Results and discussions

### 4.1 The inlet temperature and flow rate

The temperature of the inlet is an important factor that affects the overall heat effect because it directly restricts the core factor of the buried tube heat transfer capacity, which is the temperature difference between the fluid medium and the surrounding solid during the heat transfer. To determine the changes in temperature between the inlet and outlet media of the buried pipe, it is necessary to simulate the conditions of constant temperature and flow rate to carry out predictive analysis. Since the heat exchange of buried pipes is the most widely used parameter in the design phase of GSHPS, it is used here as the evaluation index of heat exchange efficiency. The calculation formula [1] is as follows.

$$Q = \rho c_p A u |t_{in} - t_{out}| \tag{4}$$

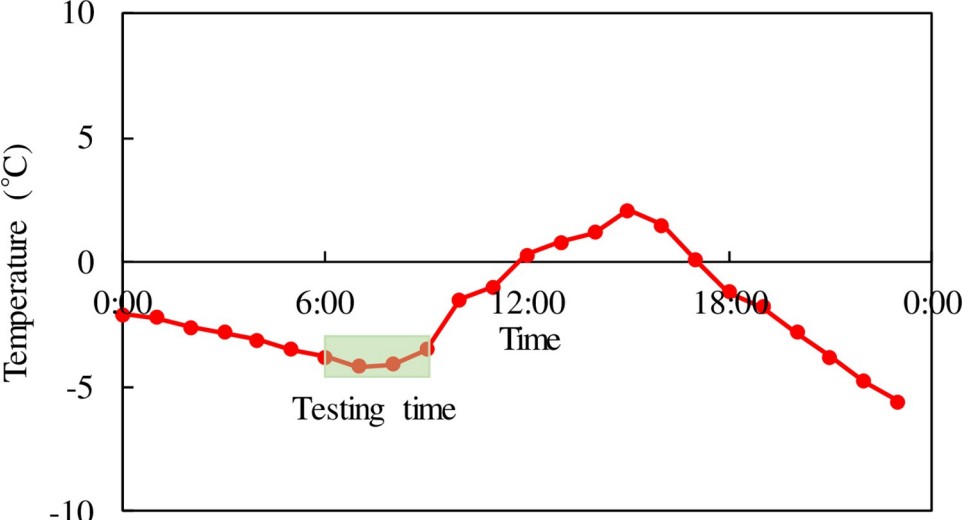

**Fig 3. Temperature characteristics of the site on the day of the experiment.**

$$q = \frac{Q}{L} \tag{5}$$

where $Q$ is the total heat exchange, kW; $q$ is the heat exchange rate per unit length, W/m; $\rho$ is the fluid density, taking the density of ethylene glycol aqueous solution as 1050 kg/m$^3$; $C_p$ is the mass-specific capacity of the fluid, 1550 J/(kg·K); $t_{in}$ is the fluid medium inlet temperature, °C; $t_{out}$ is the outlet temperature, °C. The coefficient of performance (*COP*) used to represent the thermal exchange efficiency of the GSHPS is typically denoted as

$$COP = \frac{Q}{W} \tag{6}$$

where, $W$ is the power consumption of the system, kW. Given the fixed power of the heat pump system, the total heat exchange serves as an indicator of the overall system's heat exchange efficiency.

Fig 4 shows the outlet temperature for the initial two days of system operation under 2~8°C inlet temperature conditions of the buried pipe. As the inlet temperature increases, the outlet temperature also increases, and this trend is consistent across different conditions. The outlet temperature changes can be roughly divided into three stages: rapid change, gentle change, and linear change stages.

Before the heat pump system starts operation, the medium stored in the buried pipe is at the same temperature as the underground temperature of 15°C. During the initial system operation, all of the stored medium will exit the outlet within the first 350 seconds. Thus, the outlet temperature will start changing after a short period, and the following data is presented and analyzed after 350 seconds. The rapid change phase occurs during the first 2 hours of operation when the outlet temperature rapidly drops, and the rate of decrease slows down as the system continues to operate. The greatest variation occurs at an inlet temperature of 2°C, with a drop of 8.84°C. When the inlet temperature is 8°C, the minimum temperature difference reaches 4.74°C.

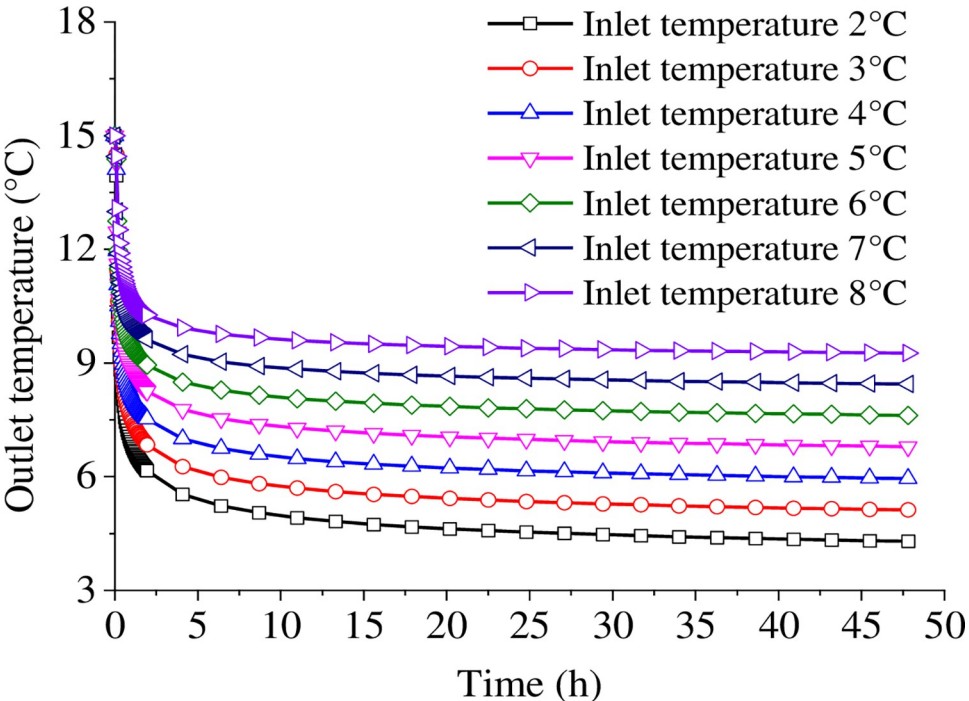

**Fig 4. Evolution of outlet temperature in two days of continuous operation.**

The temperature change in the second stage is much slower, occurring within 2~10 hours of continuous heat exchange. The temperature difference of the entire stage decreases slightly with the increase of the inlet temperature. The outlet temperature decreases at a constant rate after 10 hours of system operation. The rate of change of the outlet temperature during the linear variation phase is constant at 0.014˚C/h for an inlet temperature of 2˚C, 0.011˚C/h for an inlet temperature of 5˚C, and 0.007˚C/h for an inlet temperature of 8˚C. It's worth mentioning that the rate of decrease throughout the stage slows down with the increase in inlet temperature.

Fig 5 illustrates the difference of total heat exchange under varying inlet temperature conditions. As the inlet temperatures remain constant, the heat exchanger power exhibits a trend consistent with the outlet temperature. The heat transfer efficiency decreases with an increase in inlet temperature. After two days, the heat transfer efficiency at an inlet temperature of 2˚C is expected to be approximately 82% higher than at 8˚C, and 28% higher than at 5˚C. This decrease in efficiency can be attributed to the reduction in the temperature gradient between the medium and the surrounding objects due to the increase in inlet temperature.

During winter snow melting operation, the inlet temperature gradually drops due to external snowfall and subsequent cold weather. Hence, it is crucial to prevent the freezing of piping caused by excessively low water temperature. Based on the aforementioned calculations, the increasing of inlet temperature cannot achieve higher heat transfer efficiency. According to the technical and construction specifications of GSHPS of China, the most suitable inlet and outlet temperature difference ranges between 5–6˚C. Based on the data of the Meteorological Bureau, the average daily temperature in the Jinan area varies between -2 and 1˚C during the three days before and after a snowfall. Therefore, a suitable temperature 5˚C for inlet is recommended.

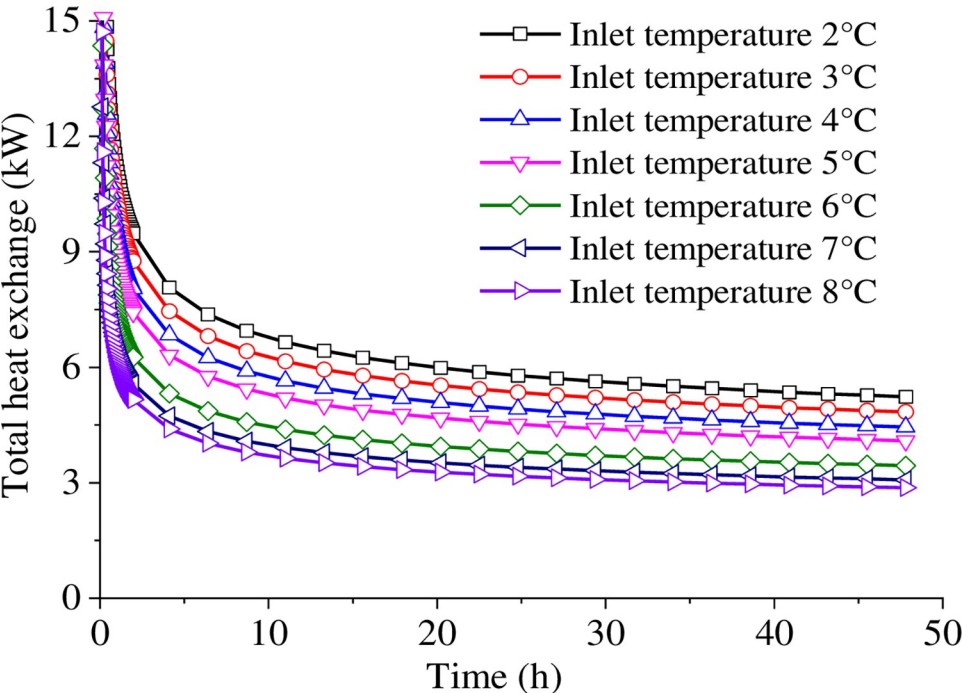

**Fig 5. Total heat exchange at different inlet temperatures.**

## 4.2 The backfill material

As the interface between the heat exchanger tube and the surrounding soil, the choice of backfill material has a significant impact on the efficiency of the heat exchanger. Referring to specifications by ASHRAE [33], Table 3 shows the thermal properties of different backfill materials commonly used in projects.

Fig 6 shows that the total heat exchange varies significantly with different backfill materials. The use of bentonite (containing 20%-30% solids) as backfill results in the lowest heat transfer efficiency throughout the operation. In contrast, the use of 10% bentonite and 90% $SiO_2$ sand as backfill results in the highest heat transfer efficiency, which is related to the larger thermal conductivity and specific heat capacity. The mixture containing 15% bentonite has a lower thermal conductivity but higher specific heat capacity than the 20% bentonite mixture, yet the latter has stronger total heat exchange. Therefore, it seems that thermal conductivity is a more important constraint on the heat exchange efficiency than specific heat capacity.

**Table 3. Thermophysical parameters of backfill materials with different content ratios.**

| Backfill materials | Thermal conductivity | Specific heat capacity |
|---|---|---|
| | W/m·K | J/kg·K |
| Bentonite | 0.73~0.75 | 325~410 |
| (Containing 20%~30% solids) | | |
| A mixture containing 20% bentonite, 80% $SiO_2$ sand | 1.47~1.64 | 870~940 |
| A mixture containing 15% bentonite, 85% $SiO_2$ sand | 1.00~1.10 | 890~1010 |
| A mixture containing 10% bentonite, 90% $SiO_2$ sand | 2.08~2.82 | 900~1100 |

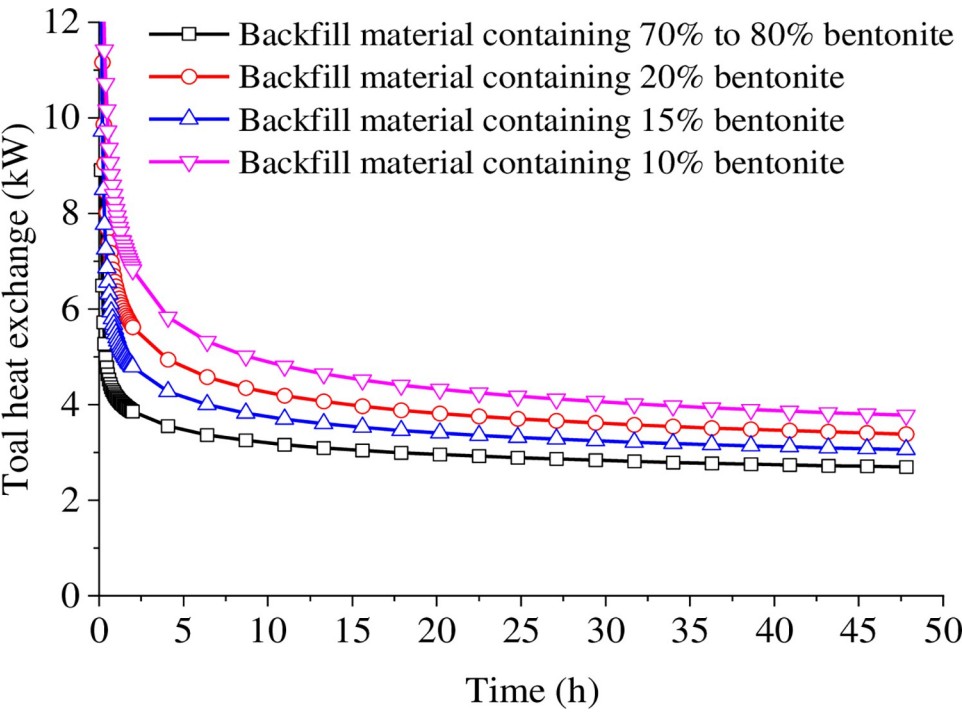

**Fig 6. Total heat exchange for different backfill materials.**

## 4.3 The depth of heat exchanger pipe

For the GHE pipe, if the design depth is too small, it may result in inadequate heat exchange, failing to meet the snow melting load requirements. Conversely, if the design depth is too large, it can lead to unnecessary drilling and backfill material cost. Therefore, the heat exchanger borehole depth is an important design parameter that must be considered. Three models with depths of 80, 100, and 120 meters were established and studied while keeping the remaining parameters and boundary conditions consistent. Fig 7 shows that the heat exchange rate per meter of the heat exchanger increases as the depth of the borehole increases. When the system runs continuously until the next day, the heat exchange rate in the 80 m depth case reaches 33.12 W/m, about 17.9% higher than that at 120 m depth. This is mainly because that increasing the depth of the buried pipe makes the flowing time in the pipe increase, and the heat extraction of each small section of the fluid medium decreases, thereby reducing the heat exchange rate per unit depth. Therefore, increasing the heat exchanger's overall design depth leads to a discounted heat exchange rate per unit depth, which must be considered during the design process.

## 4.4 Effect of the operation time

Fig 8 shows the evolution of temperature at the interface between the backfill material and the soil, that is the outside of the drilling hole (distance $r$ = 75 mm from the center point). It can be seen the temperature varies small at different depths (within 0.3°C) for any operating time. However, it is worth noting that the temperature gradually returns to the initial soil temperature ($T_{\text{ground}}$ = 15°C) at the bottom of the heat exchanger (U-joint connection position). Therefore, it can be determined that the temperature outside the backfill area is uniformly distributed with depth and with time. It also can be seen temperature dropped from 15°C to

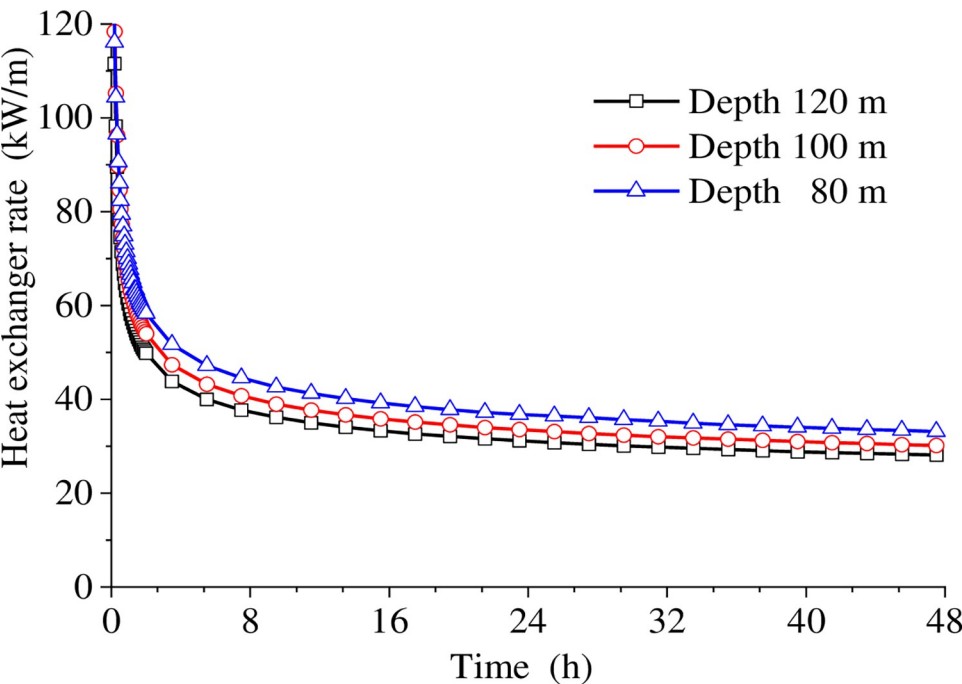

**Fig 7. Heat exchange rate per linear meter at different depths.**

about 9˚C within the 1 day of operation. For comparison, the field test results on temperature distribution of the backfill material over time are also presented. The temperature of soil layers at 2 m and 7 m from the ground was notably affected by the external atmospheric temperature, resulting in marked differences in the measured temperature of the backfill material compared to the overall curve. The initial temperature of the layers below 10 m was approximately 15˚C. As the test time for heating the bridge deck increased, the temperature of the backfill material gradually decreased. After 3 hours of testing, the overall backfill material temperature lowered to around 10˚C. The measured temperature is slightly higher than the numerical simulation results because the measurement point is located inside the borehole, while the numerical simulation is at the interface between the backfill material and the ground. In addition, the temperature variation of the backfill material at the bottom of the buried pipe was greater than that at the top.

Fig 9 shows the soil temperature variation at the depth 60 m in the radial at different operating moments. The diffusion of the temperature field in the radial with time can be seen, and the diffusion of the temperature change is about 0.5 m at 1 hour of operation. After that, the range of variation reached 0.9, 1.2, 1.8, 2.5, and 3.0 m at 12 h, 1 day, 2 days, 4 days, and 7 days of operation, respectively. Therefore, it can be concluded that the range of temperature variation gradually expands due to the appearance and increase of the soil temperature gradient from near and far during the heat exchange process. Because the heat transfer efficiency is gradually reduced throughout the process, the diffusion rate of the temperature change range decreases gradually. When the continuous operation reaches 30 days, the temperature difference between different locations is less than 0.2˚C compared to the 20th day, which tends to the equilibrium state.

Running for up to 1 hour, the soil temperature at -0.09 m on the inlet side is 0.7˚C lower than at 0.09 m on the outlet side, while the temperature difference between the two sides at ±0.2 m is 0.1˚C. This proves that at the same time, the soil temperature on the inlet side of the

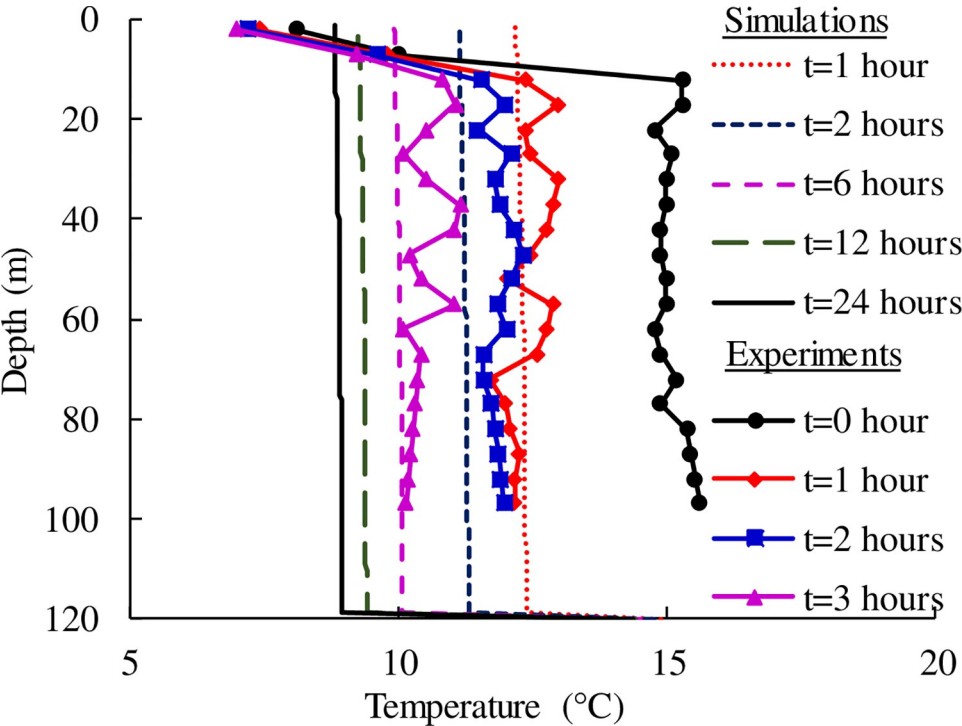

**Fig 8. Temperature change at the interface between the backfill area and the soil.**

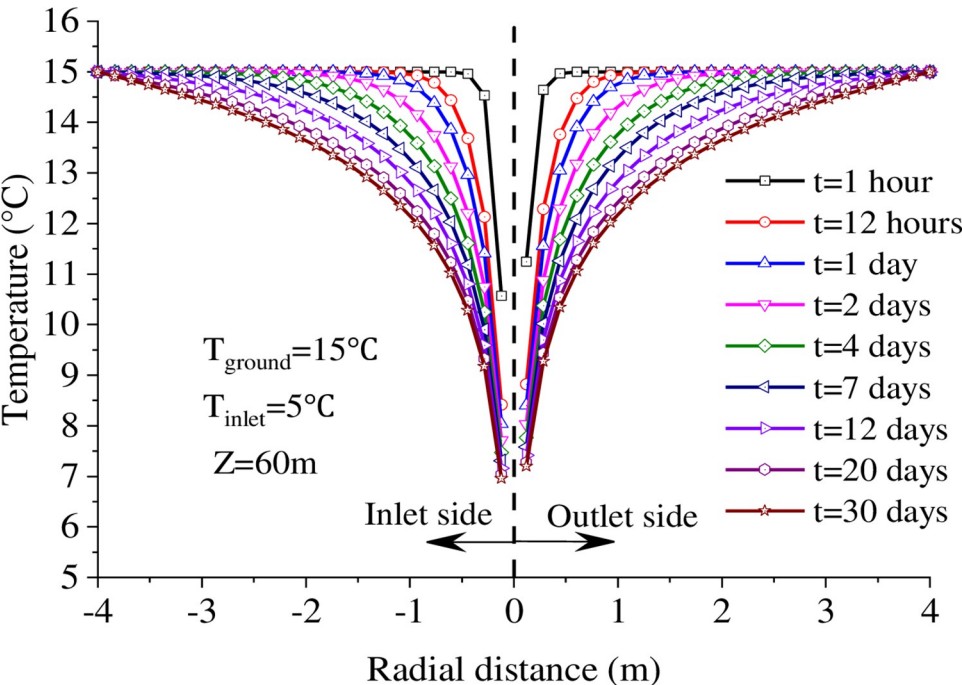

**Fig 9. Temperature curve changes in the radial direction.**

fluid is lower than the temperature at the same distance from the outlet side, and the temperature difference decreases as the distance from the heat exchanger increases. The reason for this is that the average temperature of the flowing medium in the tube on the inlet side is lower than on the outlet side during the flowing heat transfer, causing a temperature difference between the two sides of the heat source, which in turn produces an unequal temperature gradient with the soil.

## 4.5 Effect of heat exchanger hole spacing

For the snow melting case, the heat required by each pump unit is extracted and provided by five underground heat exchangers. The location of borehole arrangement of the actual design scheme is shown in Fig 1. To study the influence of hole spacing on the heat transfer characteristics, a heat transfer unit consisting of a cluster of holes was simulated, and three-dimensional modeling of hole spacing of 3m, 4m, and 5m was established, respectively, while keeping other conditions consistent. Meanwhile, the equivalent diameter method is adopted to equate the U-shaped buried pipe part to a single pipe of equivalent diameter [34]. The equivalence formula is as follows.

$$d_{ep} = (2d_{po}D_{U})^{1/2} \qquad (7)$$

where: $d_{eq}$ is the equivalent pipe diameter, m; $d_{po}$ is the outside diameter of the U-shaped buried pipe, m; $D_U$ is the U-shaped pipe foot spacing, m.

Generally speaking, the operation of GSHPS for the bridge snow melting condition is only a few days. To highlight the effects of different borehole spacing on heat exchange efficiency, the continuous operation time was extended to 45 days. The temperature field between heat exchangers in the middle depth section of the model was observed, and the total heat exchange capacity was compared to determine the optimal borehole spacing.

Fig 10 show the temperature distribution under three spacing conditions on the 45th day of continuous heat extraction. It can be observed that the average soil temperature at a spacing of 5m is the highest, indicating a weak thermal interference phenomenon, where the 13˚C temperature isotherms spread to a location centered on the heat exchanger with a diameter of about 2m. At a spacing of 4m, the average surrounding temperature is the second highest, and each heat exchanger around the 13˚C isotherms appear to intermingle with each other. Since the thermal disturbance phenomenon occurs earlier at a spacing of 3 m, the average soil temperature is lowest, and the 13˚C isotherm has intersected and spread to the outside of the heat exchanger group. The temperature between the two heat exchangers is less than 12˚C. The difference in temperature distribution precisely explains the superiority and inferiority of the heat transfer capacity under the three conditions of hole spacing.

Fig 11 shows the total heat exchange of five heat exchangers at 3, 4, and 5m spacing changes over time. It is evident that total heat exchange increases with increasing heat exchanger hole spacing. When the spacing is 3 m, thermal interference among the temperature fields around the heat exchangers occurs faster, which reduces the temperature gradient between the heat exchangers and the soil and affects heat exchange efficiency. At 2 h of operation, heat exchange under 4m spacing reaches 36.19 kW, which is 6.5% higher than under 3 m spacing. At the same time, heat exchange under 5m spacing is 37.15 kW, 2.7% higher than under 4m spacing. At 8 h, heat exchange increases by 5.4% under 4 m spacing compared to 3 m spacing, while it increases by 1.9% under 5 m spacing compared to 4 m spacing. Therefore, from the perspective of effective utilization of the ground, the optimal borehole spacing of 4 m will be better in the project.

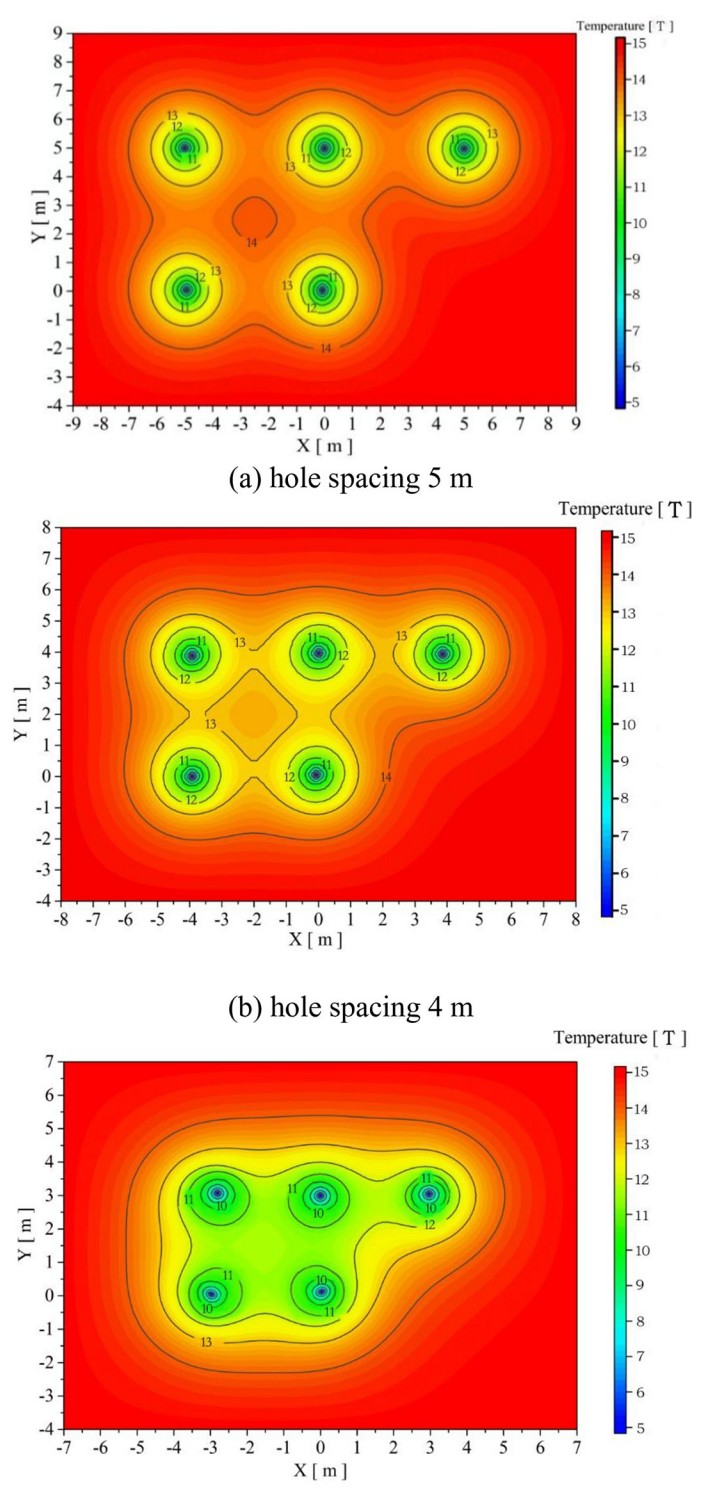

(a) hole spacing 5 m

(b) hole spacing 4 m

(c) hole spacing 3 m

**Fig 10.** Temperature field under different hole spacing for operation up to 45 days; (a) hole spacing 5m; (b) hole spacing 4m; (c) hole spacing 3m.

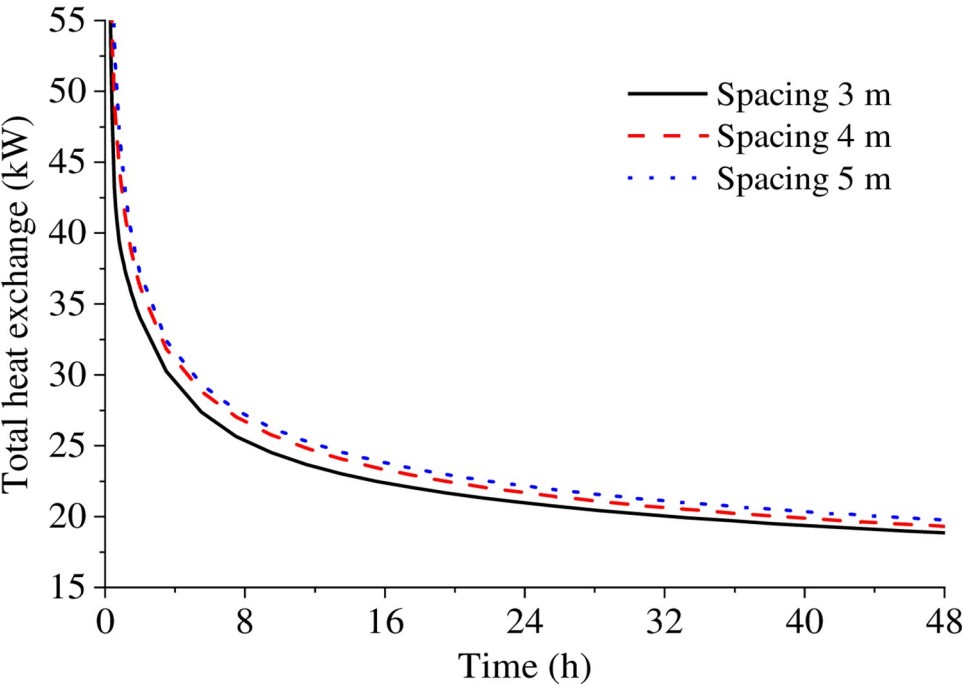

**Fig 11. Heat exchange rate under different spacing conditions.**

## 4.6 Effect of pipe spacing

Fig 12 illustrates the variation of surface temperature directly above the pipeline over time. It can be observed that as heating time progressed, the surface temperature of the road increased continuously for different pipe spacing. However, with an increase in pipe spacing, the average temperature on the road surface gradually decreased. The impact of pipe spacing between 0.15 m and 0.20 m exhibited a significant change in amplitude, while there was a substantial decrease in amplitude when the pipe spacing exceeded 0.20 m. This indicates that the influence of pipe spacing on the road surface's temperature field becomes relatively gentle as it extends to a certain distance, and the extent of impact on the temperature of the road surface's structural layer rapidly diminishes.

## 5 Conclusions

The performance of an underground heat exchanger system in regards to bridge snow melting was systematically studied, and the effects of key factors on the heat extraction capacity of the ground heat exchanger, as well as the temperature field distribution in the ground between multiple heat exchangers under various spacing conditions, were observed. The following conclusions were drawn:

1. The heat exchanger continues to extract heat throughout the process, and there are three main stages of export temperature: rapid change, gentle change, and linear change. A mixture with a smaller percentage of bentonite content and a higher percentage of $SiO_2$ sand content as backfill material benefits the heat exchange in the heat exchanger. Increasing the depth of the heat exchanger will decrease the heat exchange rate per linear meter instead of increasing it.

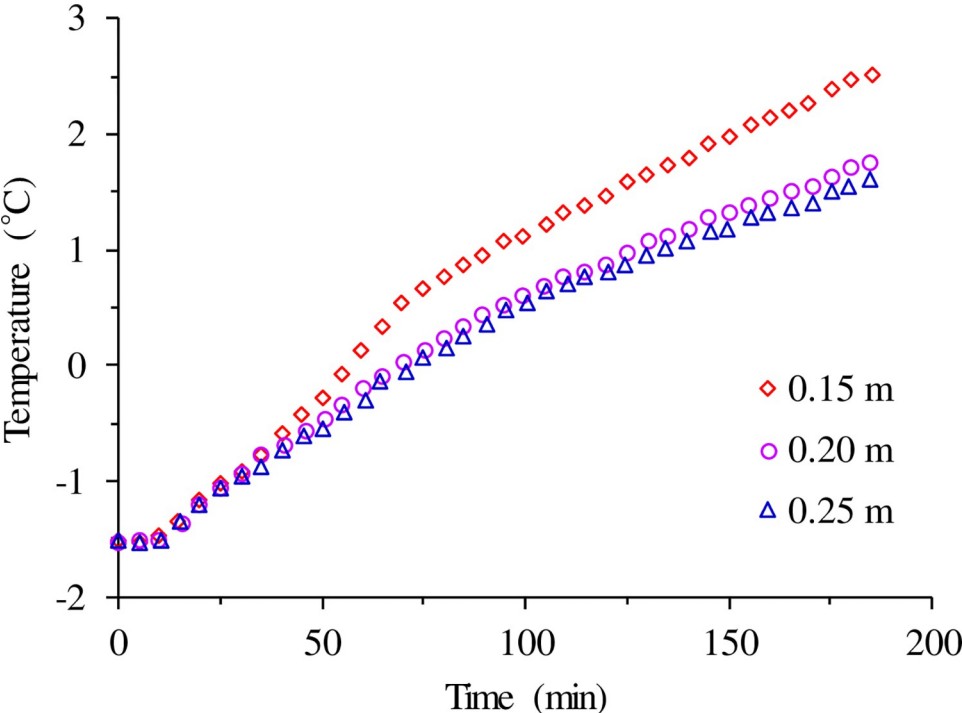

**Fig 12. The effect of spacing on bridge deck temperature.**

2. As the heat exchanger spacing expands, the thermal interference effect generated between them gradually diminishes, and the average soil temperature increases, thus enhancing the heat extraction capacity of the heat exchanger. As the operation time progresses, the temperature of the backfill material gradually decreases, and for a 3-hour operation time, the temperature changes around 5˚C.

3. The spacing of GHEs plays a pivotal role in determining the temperature gradient between the heat exchangers and the soil, thereby impacting heat exchange efficiency. In the context of the Jinan project in China, a recommended borehole spacing of 4 m is advised. Additionally, the density of buried pipes laid on the bridge deck has a notable effect on the evolution of the bridge surface temperature. A critical spacing of 0.2 m for laying buried pipes on the bridge deck is identified; beyond this value, the surface temperature shows no significant sensitivity to the spacing.

## Supporting information

**S1 File.**
(DOCX)

## Acknowledgments

The authors appreciate Jinan Department of Transportation providing the testing site.

## Author Contributions

**Conceptualization:** Xuegui Zheng.

**Data curation:** Zhongbo Song.

**Investigation:** Zhongbo Song.

**Methodology:** Xuegui Zheng.

**Validation:** Yanping Ding.

**Writing – original draft:** Xuegui Zheng.

**Writing – review & editing:** Yanping Ding.

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
