## [Decision Letter · Decision Letter 0]

28 Sep 2023

PONE-D-23-29395Heat exchange characteristics of underground and pavement buried pipes for bridge deck heating conditionsPLOS ONE

Dear Dr. ZHENG,

Thank you for submitting your manuscript to PLOS ONE. After careful consideration, we feel that it has merit but does not fully meet PLOS ONE’s publication criteria as it currently stands. Therefore, we invite you to submit a revised version of the manuscript that addresses the points raised during the review process.

We look forward to receiving your revised manuscript.

Kind regards,

S. M. Anas, Ph. D. (Structural Engg.), M. Tech

Academic Editor

PLOS ONE

Journal Requirements:

   "The first author thanks for funding from Chongqing Vocational Institute of Engineering，"

6. We note that Figure 12 includes an image of a participant in the study. 

Additional Editor Comments:

Dear Authors:

The manuscript entitled "Heat exchange characteristics of underground and pavement buried pipes for bridge deck heating conditions" [PONE-D-23-29395] was reviewed by two experts in the concerned research field. Both of them gave "Major Revision" recommendation and suggested to improve the quality of the manuscript significantly.

This editor has decided to take "Major Revision" decision on this manuscript and suggests the authors to incorporate the changes given by the reviewers carefully, and address them properly.

Thank you for your time.

Sincerely regards,

Dr. S. M. Anas

B. Tech (Civil Engg.), M. Tech (Earthquake Engg.), Ph.D. (Structural Engg.)

Academic Editor, PLoSONE

Reviewers' comments:

Reviewer's Responses to Questions

**Comments to the Author**

1. Is the manuscript technically sound, and do the data support the conclusions?

Reviewer #1: Partly

Reviewer #2: Yes

2. Has the statistical analysis been performed appropriately and rigorously? 

Reviewer #1: No

Reviewer #2: Yes

3. Have the authors made all data underlying the findings in their manuscript fully available?

Reviewer #1: No

Reviewer #2: Yes

4. Is the manuscript presented in an intelligible fashion and written in standard English?

Reviewer #1: Yes

Reviewer #2: Yes

5. Review Comments to the Author

Reviewer #1: 1. Abstract:

After reading the abstract of the article, I find that there is no clear understanding of why you conducted this research. A good abstract should include the scientific problem/challenge, which I don't see in the first 2-3 sentences of your abstract. Please, explore the abstracts of several good articles - they usually consist of approximately 30% of Background, 30% of Methodology, and 40% of Key Findings. I want to emphasize that by "key findings," I mean the key results of your study that provide valuable knowledge to the scientific community, along with the main reasons why GSHPS yielded these particular results. As it stands, your abstract has not captured the reader's interest.

So far, in your Abstract I see only a methodology that does not reflect the specifics.

2. Introduction:

In the Introduction section, the impression is given that the GSHPS field has already been thoroughly studied and holds little scientific interest. I strongly recommend that the authors clarify the limitations of previous research in this area to highlight the significance of their own work.

The authors rely on literature from 2007-2012 to describe the 'latest scientific advancements,' but it is my understanding that many developments have occurred in the field in recent years. I strongly suggest that the authors consult more recent literature to provide a more current context for their work. For example, what about phase change materials that are used in soil for thermal stabilization? Or thermosyphons? A comprehensive literature review is advised.

3. “Jinan (116°E, 37°N) is located in the North region of China, and there are several snowfalls every year” – Please be specific. Provide data on the amount of snowfalls, rather than simply stating that there is 'several.'

4. Part “2.1”. The authors constantly talk about GSHPS, but the reader still does not understand what it is. It is recommended to include a technical description.

5. Part “2.3 The inlet temperature and flow rate” – Why do the authors not use references for the literature? I am certain that the formulas are not the authors' own development.

6. On some figures, the authors use units of temperature in Celsius, while on others, they use Kelvin. Please standardize the units of measurement for consistency.

7. Figure 12 does not provide any useful information to the engineering and scientific community and is recommended for removal. What information do photos 12(b) and 12(e) convey? The reader only sees that you have some sort of sensor. If the authors believe this information is essential for understanding the paper, a wiring diagram for the sensors is recommended instead of a photo of a twisted wire.

8. Figure 15 – To save on printing costs and reduce carbon footprint, many readers print figures in black and white. I cannot analyze this graph effectively because the authors did not use different symbols to distinguish between the various lines.

9. It is puzzling that the paper completely lacks a Discussion and Results section. This article is missing one-third of the essential information. How do you intend to publish a paper without any discussion of your results?

Reviewer #2: Comments to the author

The paper provides a clear overview on the heat exchange characteristics of underground and pavement buried pipes for bridge deck heating conditions. However, the paper sets several drawbacks which should be incorporated before final publication

Abstract: Revise the abstract carefully. Which finite element software has been employed in the present study? Mention the findings of the study.

The abstract mentions “The influences of the inlet temperature, backfill material, drilling depth, and hole spacing on the heat transfer efficiency of the ground heat exchanger were studied under the alternative engineering conditions”. What are the alternative engineering conditions mentioned here? Explain them.

Keywords: Arrange the keywords in alphabetical order.

Introduction: Elaborate the introductory part.

It is written that “Many factors affect the thermal performance of ground heat exchanger (GHE), so several studies have been conducted from different perspectives”. What are those factors? Explain them.

Mention objectives and research significance of the study.

What is novelty in the present study?

Figure 2 shows Three-dimensional geometric model and the meshing of buried U-type pipe. How the meshing was achieved?

For explaining discretization and meshing of pipeline, it’s good to cite these very recently published papers in this study.

Blast performance enhancement prediction of circular column with helical mesh reinforcement and strengthened with UHPFRC plaster, and CFRP wrapping under close-in blast; R Tahzeeb, M Alam, SM Muddassir; International journal of structural engineering, 1-38.

Dynamic response of CFST column with in-plane cross reinforcement and partial CFRP wrapping upon contact blast. R Tahzeeb, M Alam, SM Anas, SM Muddassir; Innovative Infrastructure Solutions 8 (9), 241

In section 2.2, it is written that “a controlled variable research method was employed,”. How this controlled variable methodology was achieved?

Check the citations of the Figures and Tables carefully.

Elaborate the Field test and result.

The result and conclusion parts are very brief. There are only 3 findings and only 1st finding is relevant. 2nd and 3rd findings are part of result, they don’t state any relevant information. The conclusion must include at least 3 findings.

6. PLOS authors have the option to publish the peer review history of their article (what does this mean?). If published, this will include your full peer review and any attached files.

Reviewer #1: No

Reviewer #2: No

---

## [Author Response · Author response to Decision Letter 0]

19 Nov 2023

We revised the manuscript according to these comments and suggestions. In general, we have tried our best to revise our manuscript. And we hope that the revised manuscript addresses these requirements. If further revision is necessary, please contact me.

---

## [Decision Letter · Decision Letter 1]

10 Dec 2023

PONE-D-23-29395R1Heat exchange characteristics of underground and pavement buried pipes for bridge deck heating conditionsPLOS ONE

Dear Dr. ZHENG,

Thank you for submitting your manuscript to PLOS ONE. After careful consideration, we feel that it has merit but does not fully meet PLOS ONE’s publication criteria as it currently stands. Therefore, we invite you to submit a revised version of the manuscript that addresses the points raised during the review process.

We look forward to receiving your revised manuscript.

Kind regards,

Dr. S. M. Anas, Ph.D.(Structural Engg.), M.Tech(Earthquake Engg.)

Academic Editor

PLOS ONE

**Additional Editor Comments:**

Dear Authors:

The revised manuscript titled "Heat exchange characteristics of underground and pavement buried pipes for bridge deck heating conditions" [PONE-D-23-29395R1] was submitted for re-review to previous two experts in the relevant research field. One reviewer declined the re-review invitation while the other recommended "Major Revision" again. Under such conditions, the third reviewer was invited to check the revised manuscript and authors' response. The third reviewer was not fully satisfied with the authors' response and suggested "Major Revision". After considering the reviewers' recommendations and conducting a preliminary analysis of the above paper, this academic editor has decided to proceed with a "Major Revision" for this submission.

Important note from this academic editor (Dr. S. M. Anas):-

I would like to bring to your attention that citing the papers suggested by the reviewers is not mandatory for your revised manuscript. It is entirely up to you whether or not you choose to include the suggested papers in your revised version. The reviewers have provided these suggestions to enhance the quality and credibility of your research, but ultimately, the decision is yours. You have the freedom to decline including any of the suggested papers in your revised manuscript if you feel they are not relevant or do not add value to your study.

I look forward to receiving your second revised version of the above manuscript.

Thank you for your time.

Sincerely yours,

Dr. S. M. Anas

(Academic Editor)

Reviewers' comments:

Reviewer's Responses to Questions

**Comments to the Author**

1. If the authors have adequately addressed your comments raised in a previous round of review and you feel that this manuscript is now acceptable for publication, you may indicate that here to bypass the “Comments to the Author” section, enter your conflict of interest statement in the “Confidential to Editor” section, and submit your "Accept" recommendation.

Reviewer #2: All comments have been addressed

Reviewer #3: (No Response)

2. Is the manuscript technically sound, and do the data support the conclusions?

Reviewer #2: Yes

Reviewer #3: Partly

3. Has the statistical analysis been performed appropriately and rigorously? 

Reviewer #2: Yes

Reviewer #3: No

4. Have the authors made all data underlying the findings in their manuscript fully available?

Reviewer #2: Yes

Reviewer #3: No

5. Is the manuscript presented in an intelligible fashion and written in standard English?

Reviewer #2: Yes

Reviewer #3: Yes

6. Review Comments to the Author

Reviewer #2: The paper requires revision before final publication.

The introductory part is very short. Elaborate it.

Apart from China, most of the parametric studies which has been shown from Fig. 4 to Fig. 8 have been already conducted. What is the advanced novelty in the present study?

It is mentioned that “Although some experimental and simulation analysis on GSHPS had been conducted, theoretical guidance on the design of GHE with the bridge deck heating conditions has not been unified.” But the designing process has not been elaborated here as well.

What are the equations used in the simulation part i.e., heat transfer efficiency at different inlet temperatures? Heat transfer efficiency with different concentrations of media? Heat extraction capacity of different backfill materials? Heat extraction capacity per linear meter at different depths?

It is mentioned that “The heat transfer process between the ground heat exchanger and the surrounding soil is a nonstationary process that involves multi-layer heat transfer media and convection heat transfer. Due to the complex geometry of the 3D buried pipe heat exchanger, certain assumptions and simplifications were made to ensure the accuracy of the established model. These assumptions include: (1) Considering the soil as a consistent and balanced void-free entity. If the soil is assumed be void free, it will act as steel. But the real media is soil, so the assumption fails here. How will you justify this statement?

(2) Assuming that all physical parameters are constant during the heat exchange process; (3) Neglecting the external weather and shallow surface heat exchange, thermal convection, and thermal radiation.” If all physical parameters will be assumed constant, how the study shows real condition?

Fig 6 shows quality concentration. What significance does it have with heat transfer quantity?

At the end of section 4.3, conclusion has been written. Then what is the meaning of repeating same statement in the conclusion part?

Reviewer #3: In this paper, the thermal design challenges associated with ground heat exchangers for bridge heating is presented. Experimental and numerical results are presented. Although the authors addressed the raised queries, the standard of the paper still needs significant revisions.

It appears that the heat exchangers are coupled with a heat pump, I suggest modifying the title to account for this. Just ‘heat exchanger’ can be confusing.

i. In the introduction, add ‘ …heat source and heat sink in heat pumps….’

ii. You still have typos, see line 4 on page 3, the sentence should start with an uppercase letter.

iii. The first paragraph addresses performance and operation costs, it does not mention anything on the high capital costs of heat pumps?

iv. Reading the introduction, I can not find the research gaps that this study intends to fill. I believe the heat transfer capacity of heat exchangers have been studied before.

v. Given the location of the project, the annual average temperature and the ground temperature of 15oC, I am not sure how severe the winter is to require such a complex system that would work a fraction of the time in a year. An economic analysis of this and alternatives should be presented.

vi. At what moisture content were the properties taken, weighted average values will be influenced by the moisture. Also, are these average values over different seasons?

vii. The load is presented in a simplistic way, the equations used to determine this can either be in the appendix or in the paper.

viii. The extraction rate depends on several factors and can change with season and over time. Also, when you write units, leave a space between the number and the unit. i.e. 80 m and not 80m, except for percent and degree C.

ix. Page 7, ‘….the minor factors that considered….’ is a type

x. The CFD methodology is lacking in may respects, it is not clear what the flow regime is, what governing equations are solved and how. The validation of the model and the mesh dependency tests.

xi. Page 10, 40 KW, should be 40 kW, unit for power! Change this even in the results section. How was the flow rate for the fluid selected?

xii. In addition to the numerical modelling procedure, the testing itself needs further explanation. What are the uncertainties in the individual sensors? Then present a detailed and expanded uncertainty analysis.

xiii. Look at the recommendations for spacing of borehole heat exchangers, I do not think the comparison is reasonable when the literature suggests a larger separation to avoid thermal interference.

xiv. I do not think section 4.2 should be in the results section, since this is your materials. You can discuss this in the section of your field tests. If you data for this, you present the increase or decrease in performance as

7. PLOS authors have the option to publish the peer review history of their article (what does this mean?). If published, this will include your full peer review and any attached files.

Reviewer #2: No

Reviewer #3: No

---

## [Author Response · Author response to Decision Letter 1]

1 Jan 2024

We revised the manuscript according to these comments and suggestions which are in the atteched file "Response to Reviewers" .

---

## [Decision Letter · Decision Letter 2]

18 Jan 2024

Heat exchange characteristics of underground and pavement buried pipes for bridge deck heating conditions

PONE-D-23-29395R2

Dear Dr. ZHENG,

We’re pleased to inform you that your manuscript has been judged scientifically suitable for publication and will be formally accepted for publication once it meets all outstanding technical requirements.

Kind regards,

Dr. S. M. Anas, Ph.D.(Structural Engg.), M.Tech(Earthquake Engg.)

Academic Editor

PLOS ONE

Additional Editor Comments (optional):

Dear Corresponding Author and Co-Authors,

I trust this email finds you well. I am writing to inform you about the decision regarding your revised manuscript entitled "Heat exchange characteristics of underground and pavement-buried pipes for bridge deck heating conditions" (Manuscript ID: PONE-D-23-29395R2) submitted to PLOS ONE.

I am pleased to inform you that your manuscript has undergone a thorough review process, and both of the previous reviewers have accepted the revised version in its current form. Their recommendations and positive feedback on the revisions made to the manuscript were crucial in reaching this decision.

Taking into consideration the reviewers' approval, I am inclined to accept your manuscript for publication in PLOS ONE. However, as per our editorial process, the final decision is subject to the approval of the editorial board.

Once the editorial board has provided their formal approval, we will proceed with the next steps in the publication process. I will keep you informed of any further developments or requirements that may arise during this final stage.

Thank you for your dedication to improving the quality of your manuscript. Your efforts and cooperation have been instrumental in reaching this positive outcome.

Should you have any questions or concerns, please feel free to reach out. We appreciate your patience and understanding.

Best regards,

Dr. S. M. Anas

Academic Editor

PLOS ONE

Reviewers' comments:

Reviewer's Responses to Questions

**Comments to the Author**

1. If the authors have adequately addressed your comments raised in a previous round of review and you feel that this manuscript is now acceptable for publication, you may indicate that here to bypass the “Comments to the Author” section, enter your conflict of interest statement in the “Confidential to Editor” section, and submit your "Accept" recommendation.

Reviewer #2: All comments have been addressed

Reviewer #3: All comments have been addressed

2. Is the manuscript technically sound, and do the data support the conclusions?

Reviewer #2: Yes

Reviewer #3: Yes

3. Has the statistical analysis been performed appropriately and rigorously? 

Reviewer #2: Yes

Reviewer #3: Yes

4. Have the authors made all data underlying the findings in their manuscript fully available?

Reviewer #2: Yes

Reviewer #3: Yes

5. Is the manuscript presented in an intelligible fashion and written in standard English?

Reviewer #2: Yes

Reviewer #3: Yes

6. Review Comments to the Author

Reviewer #2: The authors have diligently addressed all the comments raised by this reviewer, demonstrating a thorough commitment to refining their work. Consequently, based on the substantial improvements made, this reviewer confidently recommends the paper for publication in PLOS ONE.

Reviewer #3: The raised queries have been addressed. The methodology has been clarified and is reasonable. The graph for ground temperature variation indicates that the model does not take into account the far-field boundary condition appropriately.

7. PLOS authors have the option to publish the peer review history of their article (what does this mean?). If published, this will include your full peer review and any attached files.

Reviewer #2: No

Reviewer #3: No

---

## [Editor Report · Acceptance letter]

15 Mar 2024

PONE-D-23-29395R2 

PLOS ONE

Dear Dr. Zheng, 

I'm pleased to inform you that your manuscript has been deemed suitable for publication in PLOS ONE. Congratulations! Your manuscript is now being handed over to our production team.

Kind regards, 

on behalf of

Dr. S. M. Anas 

Academic Editor

PLOS ONE